# Deletion of KLF10 Leads to Stress-Induced Liver Fibrosis upon High Sucrose Feeding

**DOI:** 10.3390/ijms22010331

**Published:** 2020-12-30

**Authors:** Junghoon Lee, Ah-Reum Oh, Hui-Young Lee, Young-Ah Moon, Ho-Jae Lee, Ji-Young Cha

**Affiliations:** 1Department of Biochemistry, Lee Gil Ya Cancer and Diabetes Institute, Gachon University College of Medicine, Incheon 21999, Korea; hun21321@naver.com (J.L.); areum1030@naver.com (A.-R.O.); hylee@gachon.ac.kr (H.-Y.L.); 2Department of Health Sciences and Technology, GAIHST, Gachon University, Incheon 21999, Korea; 3Department of Molecular Medicine, Inha University School of Medicine, Incheon 22212, Korea; yamoon15@inha.ac.kr; 4Gachon Medical Research Institute, Gachon University Gil Medical Center, Incheon 21565, Korea

**Keywords:** KLF10, NAFLD, NASH, sucrose diet, ER stress, hepatocyte cell death, fibrosis, Smad3

## Abstract

Liver fibrosis is a consequence of chronic liver injury associated with chronic viral infection, alcohol abuse, and nonalcoholic fatty liver. The evidence from clinical and animal studies indicates that transforming growth factor-β (TGF-β) signaling is associated with the development of liver fibrosis. Krüppel-like factor 10 (KLF10) is a transcription factor that plays a significant role in TGF-β-mediated cell growth, apoptosis, and differentiation. In recent studies, it has been reported to be associated with glucose homeostasis and insulin resistance. In the present study, we investigated the role of KLF10 in the progression of liver disease upon a high-sucrose diet (HSD) in mice. Wild type (WT) and *Klf*10 knockout (KO) mice were fed either a control chow diet or HSD (50% sucrose) for eight weeks. *Klf*10 KO mice exhibited significant hepatic steatosis, inflammation, and liver injury upon HSD feeding, whereas the WT mice exhibited mild hepatic steatosis with no apparent liver injury. The livers of HSD-fed *Klf*10 KO mice demonstrated significantly increased endoplasmic reticulum stress, oxidative stress, and proinflammatory cytokines. *Klf*10 deletion led to the development of sucrose-induced hepatocyte cell death both in vivo and in vitro. Moreover, it significantly increased fibrogenic gene expression and collagen accumulation in the liver. Increased liver fibrosis was accompanied by increased phosphorylation and nuclear localization of Smad3. Here, we demonstrate that HSD-fed mice develop a severe liver injury in the absence of KLF10 due to the hyperactivation of the endoplasmic reticulum stress response and CCAAT/enhance-binding protein homologous protein (CHOP)-mediated apoptosis of hepatocytes. The current study suggests that KLF10 plays a protective role against the progression of hepatic steatosis into liver fibrosis in a lipogenic state.

## 1. Introduction

Nonalcoholic fatty liver diseases (NAFLD), including nonalcoholic steatohepatitis (NASH), are frequent comorbidities that develop in patients with obesity, diabetes, hypertension, and dyslipidemia [1,2,3]. NAFLD represents a spectrum of disorders encompassing hepatic steatosis, which is a benign condition, to nonalcoholic steatohepatitis (NASH), which can cause fibrosis, cirrhosis, and hepatocellular carcinoma (HCC) [4,5,6]. During liver injury, transforming growth factor-beta (TGF-β), produced and secreted by inflammatory cells such as liver macrophages or Kupffer cells and platelets, activates hepatic stellate cells to promote fibrosis [7,8]. When chronic damage and inflammation persist, NASH progresses to liver fibrosis, with accumulation of scar tissue in the liver, thus limiting the ability of the liver to function and repair itself [9,10]. This could eventually lead to cirrhosis or hepatocellular carcinoma.

Fatty acids used in the synthesis of hepatic triglycerides (TG) come from various sources, including diet, fatty acids liberated via adipose tissue lipolysis, and de novo lipogenesis. An increase in fatty acids and reduced TG export as very-low-density lipoprotein (VLDL) or inefficient fatty acid oxidation can cause fat accumulation in the liver [11,12]. Dietary sugar is a primary stimulus for de novo hepatic lipogenesis, which may also contribute to fat accumulation in the liver [13]. The excess free fatty acids in the liver cells can induce reactive oxygen species (ROS) production, which causes lipotoxic stress in the mitochondria and endoplasmic reticulum (ER). This results in mitochondrial dysfunction, ER stress, and increased apoptotic activity [14,15].

Krüppel-like factor 10 (KLF10) is a zinc finger-containing transcription factor that is initially identified as the TGF-β-inducible early gene 1 (TIEG1) [16]. It plays an important role in TGF-β-mediated cell growth, differentiation, and apoptosis [17,18,19,20,21]. KLF10 is ubiquitously expressed, with high expression levels in muscle, pancreas, and liver. In the mouse liver, KLF10 is a circadian clock-controlled transcription factor that regulates genes implicated in glucose and lipid metabolism [22]. Moreover, the levels of KLF10 are increased in primary mouse hepatocytes upon glucose treatment by the carbohydrate response element-binding protein (ChREBP), which is another major transcription factor that regulates various genes involved in glucose and lipid metabolism [23,24,25,26]. The levels of KLF10 are increased in the mouse models of diabetes and obesity, and its gene variants exhibit a mild correlation with the development of type 2 diabetes [27,28]. KLF10 can also regulate hepatic glucose metabolism by directly regulating the expression of phosphoenolpyruvate carboxykinase, a rate-limiting enzyme of gluconeogenesis [22,27]. Previous studies suggested that KLF10 may regulate liver pathophysiology in a context-dependent manner [22,27]. However, whether KLF10 can regulate the progression of NAFLD in response to sugar intake has not yet been explored.

TGF-β is a potent and ubiquitous pro-fibrogenic cytokine that is increased in human patients with chronic liver diseases and in diverse animal models of liver fibrosis [7,8]. It transduces the signal through the Smad-dependent canonical and Smad-independent noncanonical signaling pathways, including c-Jun-N-terminal kinase (JNK), extracellular signal-regulated kinase (ERK), and p38 mitogen-activated protein kinase (MAPK). A recent publication has revealed the involvement of KLF10 in tissue fibrosis as one of the downstream effectors of TGF-β in tissue fibrosis. *Klf*10 deletion in mice led to the development of fibrosis in the skeletal muscle and heart [29,30]. These findings motivated us to study KLF10 as a target of ChREBP and TGF-β in the development of NASH during increased lipogenesis.

In this study, we investigated the role of KLF10 in the progression of NAFLD using *Klf*10 knockout (KO) mice. *Klf*10 KO mice developed hepatic fibrosis due to the induction of ER stress, inflammation, and Smad3 signaling in a lipogenic condition induced by a high-sucrose diet (HSD).

## 2. Results

### 2.1. Klf10 Deletion Induces High-Sucrose Diet-Mediated Liver Injury

A long-term HSD induces lipogenesis and hepatic TG accumulation [13]. The role of KLF10 in the development of NAFLD was evaluated in 8-week-old male WT and *Klf*10 KO mice fed either a control chow diet (CD, 10% sucrose, 50% starch) or HSD (50% sucrose, 10% starch) for eight weeks. While the WT mice maintained a constant body weight on both diets, *Klf*10 KO mice lost weight significantly upon HSD compared with CD feeding (Figure 1A). The liver weights of *Klf*10 KO mice fed an HSD were significantly lower than those of *Klf*10 KO mice fed a CD, as well as those of the WT mice (Figure 1B). As presented in Figure 1C, the bodyweight loss in HSD-fed *Klf*10 KO mice was not a result of decreased food intake. Histologically, the hepatocytes of the livers of *Klf*10 KO mice fed an HSD contained more lipid droplets compared with WT mice, which is consistent with the increased liver TG and cholesterol contents (Figure 1D,E). The livers of *Klf*10 KO mice fed an HSD presented severe damage characterized by hepatocyte ballooning and inflammatory cell infiltration (Figure 1D,F). Consistent with the histological features, the markers of liver injury, such as alanine aminotransferase (ALT) and aspartate aminotransferase (AST), were significantly increased in the plasma of HSD-fed *Klf*10 KO mice (Figure 1G).

### 2.2. Klf10 Deletion Impairs Triglyceride Packaging and Secretion upon HSD Feeding

To elucidate the reasons behind fat accumulation in the liver upon HSD feeding of *Klf*10 KO mice, we evaluated the expression of genes related to lipid synthesis, uptake, and secretion in the liver. An HSD is known to induce hepatic lipogenesis via the expression of enzymes involved in the lipogenic pathway [11]. Fatty acid synthase (FAS), which encodes a rate-limiting enzyme of fatty acid synthesis, was induced by HSD in the WT mice but not in *Klf*10 KO mice (Figure 2A). Conversely, CD36, a scavenger receptor of fatty acid uptake, was significantly increased in HSD-fed *Klf*10 KO mice, whereas CD36 in HSD-fed WT mice was decreased compared with that of the CD-fed WT mice (Figure 2A). The genes involved in VLDL packaging and secretion, such as microsomal TG transfer protein (*Mttp*) and apolipoprotein B (*Apob*), were significantly reduced in the livers of HSD-fed *Klf*10 KO mice (Figure 2B). Consistent with these results, plasma TG, especially cholesterol, was significantly decreased in HSD-fed *Klf*10 KO mice compared with the WT mice (Figure 2C). To determine whether the decreased plasma TG and cholesterol resulted from the VLDL secretion from liver, we injected the nonionic surfactant poloxamer 407 (a lipoprotein lipase inhibitor) to inhibit TG hydrolysis and measured the entry of TG into plasma [31]. CD had no effect on the plasma TG levels in either WT or *Klf*10 KO mice. However, HSD significantly decreased the plasma TG levels in *Klf*10 KO mice compared with WT mice (Figure 2D). These results indicate that enhanced lipid uptake and decreased VLDL packaging and secretion may play roles in lipid accumulation in *Klf*10 KO mice. Fatty acid synthesis is unlikely to contribute to fat accumulation in HSD-fed *Klf*10 KO mice, as the expression of FAS was actually decreased in these mice.

### 2.3. Klf10 Deletion Induces Oxidative Stress and ER Stress upon HSD Feeding

Impaired lipid secretion by MTTP inhibition was shown to increase free cholesterol in the mitochondria and ER, which resulted in oxidative stress and ER stress in the liver [32]. Therefore, we first measured the oxidative stress-related gene expression. The mRNA and protein levels of nuclear factor erythroid-derived 2-like 2 (*Nrf*2), NAD(P)H dehydrogenase quinone 1 (*Nqo*1), and sulfiredoxin 1 (*Srxn*1) were significantly increased in the livers of HSD-fed *Klf*10 KO mice compared with those of the WT mice (Figure 3A,B). The enzymes superoxide dismutase 2 (*Sod*2) and *Catalase*, which are involved in antioxidant defense, were significantly decreased in HSD-fed *Klf*10 KO mice (Figure 3A). 8-oxo-2′-deoxyguanosine (8-oxo-dG), a biomarker of oxidative DNA damage, was higher in *Klf*10 KO mice than in the WT mice upon HSD feeding (Figure 3C). These results suggest that *Klf*10 deletion induced oxidative DNA damage in HSD possibly via impaired detoxification of ROS.

We also measured the levels of known ER stress markers. The livers of HSD-fed *Klf*10 KO mice exhibited increased phosphorylation of protein kinase R-like ER kinase (PERK) and eukaryotic translation initiation factor 2α (eIF2α) as well as C/EBP homologous protein (CHOP, a key component in ER stress-mediated apoptosis), which were not observed in WT mice (Figure 4A). As expected, the mRNA levels of eIF2α downstream target genes, such as *Atf*3, *Chop*, *Gadd*34, *Noxa*, and *Puma*, were significantly increased in the livers of HSD-fed *Klf*10 KO mice (Figure 4B). These results suggest that increased activation of PERK/CHOP signaling in the livers of *Klf*10 KO mice was responsible for the liver injury upon HSD feeding.

### 2.4. Klf10 Deletion Increases Stress-Mediated Hepatic Cell Death upon HSD Feeding

Recent findings have demonstrated that oxidative stress and ER stress are closely associated with inflammation and hepatocyte cell death [33,34]. Thus, we determined the levels of the genes involved in inflammation and cell death in our experimental system. The mRNA levels of proinflammatory cytokines, such as tumor necrosis factor α (*Tnfα*) and monocyte chemoattractant protein 1 (*Mcp*1), were significantly increased in the livers of HSD-fed *Klf*10 KO mice (Figure 5A). Moreover, stress-activated JNK and ERK1/2 MAP kinases were activated in the livers of HSD-fed *Klf*10 KO mice (Figure 5B). Then, we determined whether the HSD-induced ER stress and oxidative stress led to liver cell death in HSD-fed *Klf*10 KO mice. The livers of HSD-fed *Klf*10 KO mice contained more TUNEL-positive cells compared with those of WT mice (Figure 5C). The effects of *Klf*10 deletion on hepatic cell death were also determined in isolated primary hepatocytes via incubation in media containing glucose or fructose. The cell survival rates of the primary hepatocytes isolated from *Klf*10 KO mice were significantly reduced by high-glucose and high-fructose feeding (Figure 5D). These results suggest that *Klf*10 deletion induced stress-mediated cell death, which may explain the decreased liver weight observed in HSD-fed *Klf*10 KO mice.

### 2.5. Klf10 Deletion Induces Liver Fibrosis via the Activation of the TGF-β- Smad3 Pathway

The increased liver inflammation and cell death in HSD-fed *Klf*10 KO mice motivated us to evaluate liver fibrosis. Hepatic fibrosis was determined by Sirius Red staining (Figure 6A). Interestingly, large quantities of collagen fibers accumulated in the livers of HSD-fed *Klf*10 KO mice, whereas fibrosis was hardly observed in HSD-fed WT mice. Histological analysis also revealed that HSD-fed *Klf*10 KO mice developed severe fibrosis (Figure 6B). We determined the mRNA and protein expression of genes involved in fibrosis, including TGF-β1, Smad3, plasminogen activator inhibitor type 1 (PAI-1), α-smooth muscle actin (α-SMA), and collagen type I alpha 1 chain (Col1a1). The expression of these genes was increased only in the livers of HSD-fed *Klf*10 KO mice (Figure 6C,D). Many previous studies have shown that TGF-β-mediated activation of Smad3 pathway plays a critical role during the process of liver fibrosis. During chronic liver injury, Smad3 is phosphorylated at the carboxyl terminal motif by TGF-β as well as phosphorylated at the linker region by MAPK pathway [35]. As shown in Figure 6C,D, the mRNA and protein levels of Smad3 were increased in HSD-fed *Klf*10 KO mice. Moreover, Smad3 phosphorylation at the linker region and carboxyl terminal motif was also significantly increased in the livers of HSD-fed *Klf*10 KO mice (Figure 6E). These results indicate that *Klf*10 deletion contributed to the activation of hepatic stellate cells, which worsened liver fibrosis in HSD-fed *Klf*10 KO mice.

## 3. Discussion

In the present study, we demonstrated that the absence of *Klf*10 leads to significantly worse liver pathology upon HSD feeding. HSD-fed *Klf*10 KO mice developed severe liver injury due to the hyperactivation of the ER stress response and CHOP-mediated apoptosis of hepatocytes. Furthermore, *Klf*10 deletion activated hepatic stellate cells mediated by TGF-β/Smad3 signaling, which resulted in NASH progression to fibrosis.

HSD induced hepatic steatosis via the induction of lipogenic genes in the liver, as demonstrated by WT mice in this study. However, it did not induce lipogenic gene expression in the livers of *Klf*10 KO mice, although the levels of TG and cholesterol in the livers were increased. Fat accumulation in the livers of *Klf*10 KO mice appeared to be a result of increased fatty acid uptake and decreased VLDL secretion, as demonstrated by the induction of the CD36 expression and decreased MTTP expression. Interestingly, hepatic steatosis and fibrosis observed in HSD-fed *Klf*10 KO mice were also reported in *Mttp* KO mice [36]. MTTP transfers TG to the newly synthesized ApoB, which is a critical step in VLDL assembly in hepatocytes. Pharmacological inhibition or genetic ablation of MTTP can result in the accumulation of cholesterol in the membranes of the mitochondria and ER, which causes ER stress and subsequent hepatocyte apoptosis and fibrosis [37,38], as observed in the livers of HSD-fed *Klf*10 KO mice. Hepatic inflammation and severe fibrosis in *Klf*10 KO mice were accompanied by a significant increase in proinflammatory cytokine expression and inflammatory cell infiltration, compared with *Mttp* KO mice. These discrepancies may be due to the difference in diet composition, i.e., a high-sucrose diet vs. a high-fat diet. The increased fatty acid import may represent another factor that induces inflammation. CD36 is a cell surface protein that enhances lipid accumulation in hepatocytes by facilitating fatty acid uptake, impairing fatty acid oxidation, which can activate the proinflammatory JNK/NF-κB signaling pathway. As the deletion of liver fatty acid binding protein (*L-Fabp*) in *Mttp* KO mice improved hepatic steatosis and fibrosis [36], increased fatty acid flux and intrahepatic cholesterol accumulation could be mediators of ER stress and fibrosis, although the mechanisms involved in this process remain to be elucidated.

Reduced *Mttp* expression is an important observation in the steatosis of HSD-fed *Klf*10 KO mice. Several transcription factors, such as small heterodimer partner (SHP), ChREBP, sterol regulatory element-binding protein (SREBP), and forkhead transcription factor1 (Foxo1), regulate *Mttp* gene transcription [39,40,41,42]. The proximal promoter region of *Mttp* also contains putative KLF10 binding sites, which strongly suggests that *Mttp* may be a direct target gene of KLF10, and the reduced *Mttp* expression in *Klf*10 KO mice could result from the reduction of *Mttp* transcription. However, we cannot rule out the possibility of a secondary effect in the damaged liver. Signal pathways responsible for controlling *Mttp* expression in response to an HSD should be investigated in the future.

We unexpectedly determined that KLF10 is required to protect hepatocytes from apoptosis and to preserve the liver function in response to sugar intake. It has been reported that transient hepatic ER stress alleviates misfolded protein stress without triggering hepatocyte cell death, whereas prolonged ER stress activates proapoptotic protein expression, inhibits prosurvival factors, and eventually triggers apoptosis via the PERK/CHOP pathway [43,44]. Our data provide evidence that hepatic ER stress dysregulation is a major cause of hepatocyte cell death in HSD-fed *Klf*10 KO mice. Sucrose feeding had no significant effect on the expression of ER stress markers in the livers of WT mice; however, it activated the proapoptotic PERK-CHOP pathway in the livers of *Klf*10 KO mice. Of note, recent data showed that *Klf*10 deficiency worsens hepatocyte death and fibrosis upon methionine and choline deficient fibrogenic diet that blocks VLDL secretion, causing oxidative stress, inflammation, and ER stress [45]. Fibroblast growth factor 21 (FGF21), which is a direct target of ChREBP, is required for normal hepatic metabolic response to fructose consumption, and the absence of FGF21 leads to liver inflammation and fibrosis in mice on a high-fructose or methionine- and choline-deficient diet [46,47]. Moreover, FGF21 is induced by ER stress in a PERK-eIF2α-ATF4-dependent manner, and its deletion accelerates ER stress-induced hepatic injury [48]. In HSD-fed *Klf*10 mice, hepatic expression of FGF21 is significantly increased (data not shown), suggesting that the hepatic injury present in *Klf*10 KO mice may not be because of the impaired FGF21 response elicited by ER stress on a HSD. These results suggest that impaired KLF10 signaling could be an underlying cause of steatosis progression to fibrosis in certain cases of human NAFLD.

The TGF-β signaling pathway plays a significant role in the activation of hepatic stellate cells in liver fibrosis. In HSD-fed *Klf*10 KO mice, we observed the activation of TGF-β/Smad signaling. In the context of hepatic fibrosis, Smad3 is known as a pro-fibrotic transcription factor [49]. Phosphorylation of the carboxyl terminal motif of Smad3 occurs via the TGF-β receptor, whereas phosphorylation of the linker region of Smad3 occurs via MAPK [35,50]. Phosphorylation of the linker region of Smad3 rather than carboxyl terminal motif seems to be dominant in the course of transdifferentiation of HSCs to myofibroblasts, which produce extracellular matrix molecules that contribute to liver fibrosis [51]. Indeed, increased Smad3 linker phosphorylation was observed in the livers of *Klf*10 KO mice. Understanding the molecular mechanism of how Smad3 is induced and the linker region is phosphorylated when KLF10 is absent is important for the identification of therapeutic targets to control the activation of hepatic stellate cells in fibrosis.

Further, there is evidence of an association between *KLF10* and human fibrotic disease. A study by the Spelsberg group revealed a positive correlation between six missense mutations in *KLF10* and hypertrophic cardiomyopathy [52]. Furthermore, these *KLF10* variants are associated with increased pituitary tumor transforming gene 1 (PTTG1) promoter activity and protein expression. PTTG1 has also been reported to be functionally required for hepatic fibrosis progression in an animal model of chronic liver injury [53]. Collectively, these data suggest that KLF10 may be a potential biomarker of hepatic fibrosis progression in NAFLD or other types of chronic liver disease.

In conclusion, this study demonstrated that KLF10 is required for the protection against the progression of hepatic steatosis to NASH with fibrosis upon HSD. Our findings highlight the need for an in-depth understanding of the regulation and effects of the KLF10 pathway upon sucrose feeding and as well as the progression of other ER stress-related diseases.

## 4. Materials and Methods

### 4.1. Mice and Diets

The wild type (WT) C57BL/6J mice were purchased from Jackson Laboratory (Bar Harbor, ME, USA). *Klf*10 KO mice were kindly provided by Dr. Woon-Kyu Lee (Inha University, Incheon, Korea) [54]. The mice were maintained on a CD containing 60% carbohydrate (~10% sucrose, PicoLab Rodent Diet 20, Orient Bio, Gyeonggi Province, Korea) under a 12-h light/dark cycle. The 8-week-old male mice were fed either a CD or HSD (50% sucrose/60% carbohydrate, D10001 AIN-76A; Research Diets, Inc., New Brunswick, NJ, USA) for eight weeks. Their body weight and food intake were measured weekly. At the end of the experiment, mice fasted for 2 h to minimize any effects of immediate feeding on gene expression and plasma biochemistry, followed by euthanization. Blood was immediately collected via the hepatic portal vein, and plasma was obtained by centrifugation at 300 × g for 15 min at 4 °C. Plasma samples were stored at −80 °C for biochemical analysis. The livers were removed, weighed, and either formalin-fixed or snap-frozen in liquid nitrogen. All animal studies were performed in accordance with the protocols approved by the Institutional Animal Care and Use Committee of the Lee Gil Ya Cancer and Diabetes Institute, Gachon University (LCDI-2019-0003).

### 4.2. Plasma and Tissue Biochemical Analysis

The levels of plasma ALT, AST, TG, cholesterol, HDL-cholesterol, and LDL-cholesterol were determined via automated analysis (Model AU-480; Olympus, Tokyo, Japan). In addition, the liver TG levels were determined as previously described [55] using the TG Quantification Kit (BioVision, Inc., Milpitas, CA, USA), and the liver cholesterol levels were determined using a Multi-Calibrator Lipid (Fujifilm Wako Pure Chemical, Tokyo, Japan).

### 4.3. Histological Evaluation

The liver samples were fixed in neutral-buffered formalin and embedded in paraffin blocks according to standard procedures. Paraffin-embedded tissue sections were incubated in xylene for 15 min. After rehydration in graded ethanol, antigen retrieval was performed in IHC-Tek^TM^ Epitope Retrieval Solution (IHC World, Ellicott City, MA, USA) by heating in a retrieval steamer for 40 min. Then, the sections were allowed to cool for 20 min. Subsequently, the sections were incubated with 3% H_2_O_2_ at room temperature for 30 min, blocked for 5 min, and then incubated with anti-8-oxo-dG (8-oxo-2′-deoxyguanosine) antibodies (Trevigen, Gaithersburg, MD, USA) overnight. The next day, the sections were incubated with the goat anti-mouse IgG HRP-conjugated secondary antibody (Merck Millipore, Temecula, CA, USA) for 30 min, washed with Dulbecco’s phosphate-buffered saline containing Tween-20, and then stained with 3,3′-diaminobenzidine. Hematoxylin and eosin and Sirius Red staining were processed to confirm the pathological analysis of liver disease by the KPNT (Korea Pathology Technical Center, Cheongju, Korea). Histological scoring (Appendix A
Appendix A) was performed as previously described [56].

### 4.4. In Vivo VLDL Secretion

Poloxamer 407 (P-407) was purchased from Sigma-Aldrich (St. Louis, MO, USA); 10% (*w*/*v*) P-407 solution was prepared using saline. Eight-week-old male WT and *Klf*10 KO mice were fed either a CD or HSD for 1-week. Then, mice were fasted for 5 h and intraperitoneally injected with P-407 (1 g/kg). Blood was collected from the tail vein before injection (0 h) and at 1 h after injection. Plasma was separated, and TG was determined.

### 4.5. RNA Isolation and Quantitative Real-Time Polymerase Chain Reaction

Total RNA was isolated from the mouse liver tissue using RNAiso Plus (Takara, Shiga, Japan). Purified total RNA was reverse-transcribed using the PrimeScript™ RT Reagent Kit with gDNA Eraser (Takara, Shiga, Japan). Gene-specific primers were designed using Primer Express Software (PerkinElmer Life Sciences, Waltham, MA, USA) and validated via analysis of template titration and dissociation curves. Quantitative real-time polymerase chain reaction (qPCR) was conducted on a CFX384 Touch™ Real-Time PCR Detection System (Bio-Rad Laboratories, Inc., Hercules, CA, USA) using SYBR^®^ Premix Ex Taq™ II, ROX Plus (Takara, Shiga, Japan). Relative gene expression was determined using the 2–ΔΔCT method [57], with the gene encoding ribosomal protein, large, P0 (*Rplp0*), which serves as the internal control. The primer sequences used are presented in Appendix A
Appendix A.

### 4.6. Preparation of Nuclear and Cytoplasmic Proteins from Cells and Tissues

Nuclear and cytoplasmic protein fractions were prepared using commercial reagents, NE-PER Nuclear and Cytoplasmic Extraction Reagents (Thermo Fisher Scientific, Waltham, MA, USA) and PRO-PREP™ Protein Extraction Solution (Intron biotechnology, Seongnam, Korea) according to the manufacturer’s instructions. Protein concentrations were determined using the Pierce™ BCA Protein Assay Kit (Thermo Fisher Scientific).

### 4.7. Western Blotting

Western blot analyses were conducted according to standard protocols. The following antibodies were used: anti-FAS, anti-Nrf2, anti-NQO1, anti-Srxn1, anti-PERK, anti-JNK, anti-TGFβ1, anti-Col1a1, anti-fibronectin, anti-α-SMA, and anti-PAI1 from Santa Cruz, (Santa Cruz, CA, USA); anti-eIF2α, anti-phospho-eIF2α (Ser51), anti-CHOP, anti-ERK, anti-phospho-ERK (Tyr204/Tyr187), anti-phospho-JNK (Thr183/Tyr185), anti-p38, anti-phospho-p38 (Thr180/Tyr182), anti-Smad3, and anti-phospho-Smad3 (Ser423/425) from Cell Signaling Technology (Beverly, MA, USA); anti-phospho-PERK (Thr980) from Thermo Fisher Scientific; anti-phospho-Smad3 (Thr179) and anti-phospho-Smad3 (Ser208) from Invitrogen (Carlsbad, CA, USA); anti-TATA-binding protein (TBP) from Abcam (Cambridge, MA, USA); and anti-glyceraldehyde-3-phosphate dehydrogenase (GAPDH) from Merck Millipore. TBP and GAPDH were used as a loading control.

### 4.8. Preparation of Mouse Hepatocytes and Cell Viability Assay

Primary hepatocytes were isolated from 8–10-week-old WT and *Klf*10 KO mice by perfusion with collagenase, as previously described [58]. The primary hepatocyte viability was evaluated by colorimetric assay using the Cell Counting Kit-8 (Dojindo Molecular Technologies, Rockville, MD, USA). The isolated hepatocytes were placed on 96-well plates at a density of 0.25 × 10^5^ cells/well and incubated in Hepatozyme (Life Technologies, Carlsbad, CA, USA) for 24 h, after which the media was changed to DMEM supplemented with 5 mM glucose, 25 mM glucose, 25 mM glucose/fructose, or 25 mM glucose with 0.5 mM palmitic acid. After 24 h, CCK-8 reagent (10 µL) was added to the wells and incubated for 3 h, followed by measurement of absorbance at 450 nm using a microplate reader.

### 4.9. TUNEL Assay

The TransDetect^®^ In Situ Fluorescein TUNEL Apoptosis Detection Kit was purchased from Transgen Biotech (Beijing, China) and used according to the manufacturer’s instructions. The fluorescein-labeled DNA was detected via fluorescence microscopy.

### 4.10. Statistical Analysis

Data are expressed as mean ± standard error of the mean. Charts were generated using GraphPad Prism 5.0 (GraphPad Software, Inc., San Diego, CA, USA). Statistical analysis was conducted using the SPSS software (Version 17.0; SPSS Inc., Chicago, IL, USA). Data were analyzed using the Mann–Whitney U test, and *p* < 0.05 was considered statistically significant.

## Figures and Tables

**Figure 1 ijms-22-00331-f001:**
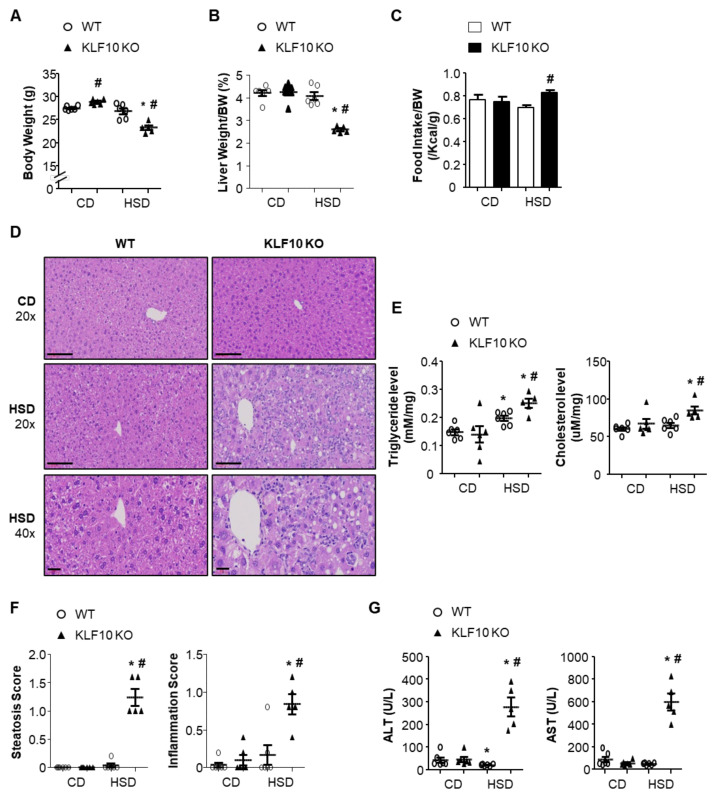
*Klf*10 KO mice develop HSD-induced liver injury. Eight-week-old WT and *Klf*10 KO mice were fed either CD or HSD for eight weeks (n = 5–6 mice/group). (**A**) Body weights and (**B**) liver weights were determined after eight weeks of either diet. (**C**) Food intake (average over seven days) was measured in WT and *Klf*10 KO mice. (**D**) Representative histological images of H&E-stained liver sections (Scale bars: 60 μm, Magnification: 20×; enlarged images in the bottom panels of D, 40×). (**E**) Triglycerides and cholesterol in the liver were measured from liver samples. (**F**) The grade of the steatosis and inflammation score was calculated. (**G**) Plasma ALT and AST levels were measured as liver injury markers. All data are representative of at least three independent experiments and expressed as mean ± SEM. Charts were produced using GraphPad Prism 5.0. Statistical differences were determined by two-way ANOVA with Mann–Whitney *U* test using SPSS v17.0. * *p* < 0.05 vs. genotype-matched, CD-fed group and # *p* < 0.05 vs. diet-matched, genotype control. WT, wild type; KO, knockout; CD, control chow diet; HSD, high-sucrose diet.

**Figure 2 ijms-22-00331-f002:**
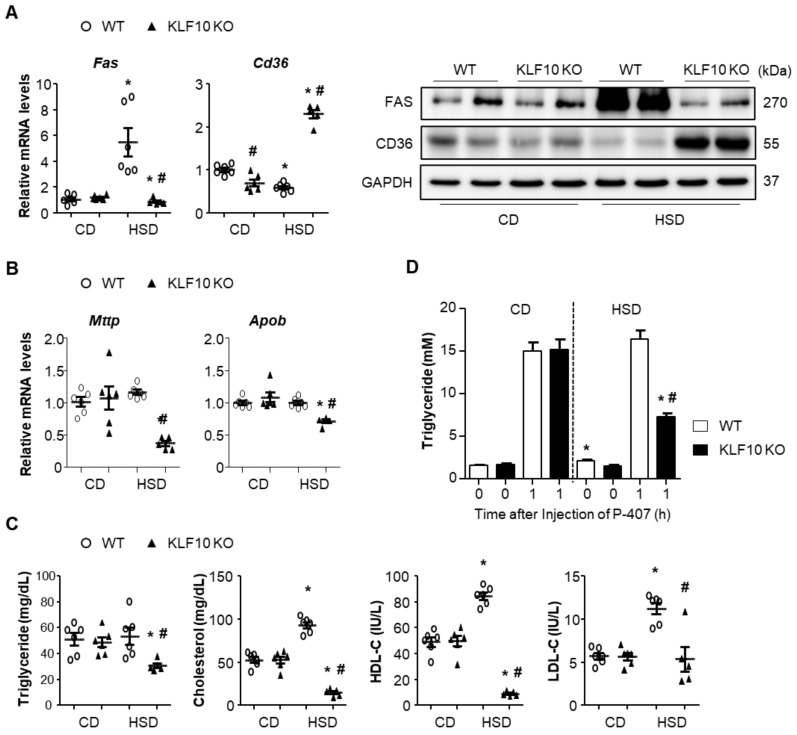
*Klf*10 deletion impairs triglyceride package and secretion upon high-sucrose feeding. Eight-week-old WT and *Klf*10 KO mice were fed either CD or HSD for eight weeks (n = 5–6 mice/group). Liver genes and proteins involved in (**A**) lipogenesis (fatty acid synthase, FAS), fatty acid uptake (fatty acid translocase, CD36), and (**B**) lipid secretion (microsomal triglyceride transfer protein, MTTP, and Apolipoprotein B, ApoB) were analyzed via qPCR and Western blotting. (**C**) Plasma triglyceride, cholesterol, HDL-cholesterol (HDL-C), and LDL-cholesterol (LDL-C) were measured. (**D**) In vivo VLDL secretion from livers of mice was determined. Eight-week-old WT and *Klf*10 KO mice were fed either CD or HSD for one week and then injected with poloxamer 407 (P-407) (n = 6 mice/group). Blood was collected at the indicated time points, and TG was measured. All data are representative of at least three independent experiments and expressed as mean ± SEM. Charts were produced using GraphPad Prism 5.0. Statistical differences were determined by two-way ANOVA with Mann–Whitney *U* test using SPSS v17.0. * *p* < 0.05 vs. genotype-matched, CD-fed group and # *p* < 0.05 vs. diet-matched, genotype control. WT, wild type; KO, knockout; CD, control chow diet; HSD, high-sucrose diet.

**Figure 3 ijms-22-00331-f003:**
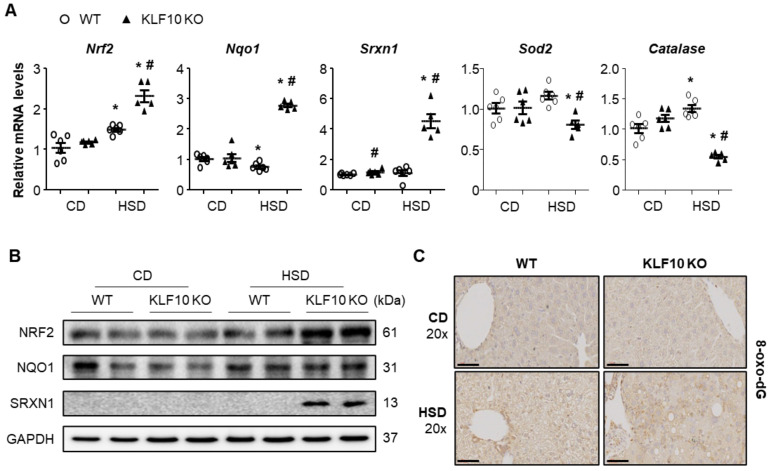
*Klf*10 deletion induces oxidative stress upon high-sucrose feeding. Eight-week-old WT and *Klf*10 KO mice were fed either CD or HSD for eight weeks (*n* = 5–6 mice/group). Changes in hepatic mRNA and protein expression involved in oxidative stress response were analyzed via (**A**) qPCR and (**B**) Western blotting. Antioxidant genes (*Nrf2*, *Nqo1*, and *Srxn1*) and ROS scavenger genes (*Catalase* and *Sod*2) were analyzed. Target gene expression was normalized to the expression of *Rplp0*. (**C**) Representative 8-oxo-2′-deoxyguanosine (8-oxo-dG)-stained livers (Scale bars: 60 μm, Magnification: 20×). All data are representative of at least three independent experiments and expressed as mean ± SEM. Charts were produced using GraphPad Prism 5.0. Statistical differences were determined by two-way ANOVA with Mann–Whitney *U* test using SPSS v17.0. * *p* < 0.05 vs. genotype-matched, CD-fed group and # *p* < 0.05 vs. diet-matched, genotype control. WT, wild type; KO, knockout; CD, control chow diet; HSD, high-sucrose diet.

**Figure 4 ijms-22-00331-f004:**
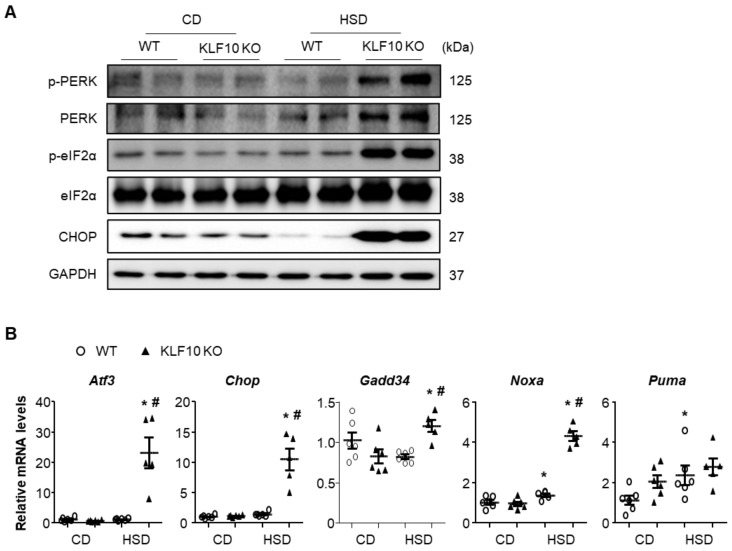
*Klf*10 deletion activates the proapoptotic PERK-CHOP pathway upon high-sucrose feeding. Eight-week-old WT and *Klf*10 KO mice were fed either CD or HSD for eight weeks (n = 5–6 mice/group). (**A**) Western blot analysis of p-PERK, PERK, p-eIF2α, eIF2α, and CHOP in the livers of WT and *Klf*10 KO mice fed either CD or HSD. (**B**) Relative expression of genes involved in ER stress (*Atf3*, *Chop*, *Gadd*34, *Noxa*, and *Puma*) in the liver was analyzed via qPCR. Target gene expression was normalized to the expression of *Rplp0*. All data are representative of at least three independent experiments and expressed as mean ± SEM. Charts were produced using GraphPad Prism 5.0. Statistical differences were determined by two-way ANOVA with Mann–Whitney *U* test using SPSS v17.0. * *p* < 0.05 vs. genotype-matched, CD-fed group and # *p* < 0.05 vs. diet-matched, genotype control. WT, wild type; KO, knockout; CD, control chow diet; HSD, high-sucrose diet.

**Figure 5 ijms-22-00331-f005:**
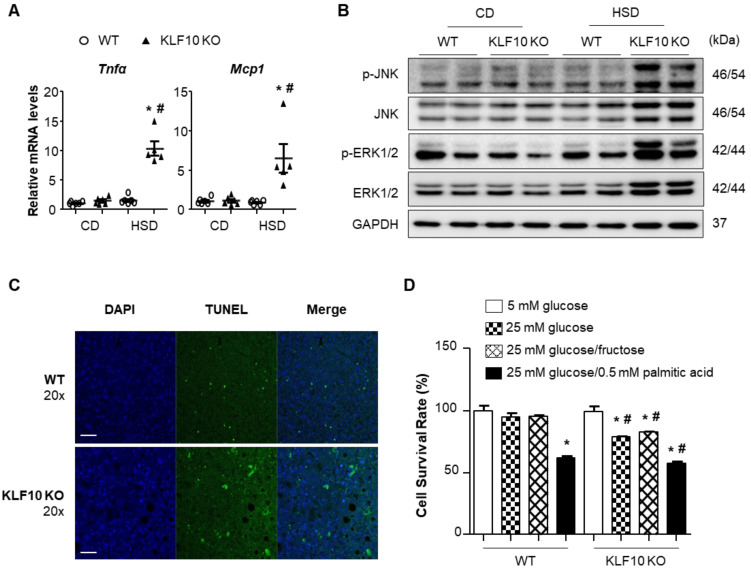
High sucrose diet activates stress-mediated inflammation and cell death in *Klf*10 KO mouse livers. Eight-week-old WT and *Klf*10 KO mice were fed either CD or HSD for eight weeks (*n* = 5–6 mice/group). (**A**) Relative mRNA levels of proinflammatory cytokines *Tnf*α and *Mcp*1 in the liver were analyzed via qPCR. * *p* < 0.05 vs. genotype-matched, CD-fed group and # *p* < 0.05 vs. diet-matched, genotype control. (**B**) Western blot analysis of p-JNK, JNK, p-ERK1/2, and ERK1/2, in the livers of WT and *Klf*10 KO mice fed either CD or HSD. (**C**) Representative images of the TUNEL assay were used to confirm the induction of apoptosis in the liver sections (Scale bars: 50 μm, Magnification: 20×). (**D**) Primary hepatocytes were isolated from WT and *Klf*10 KO mice. The CCK-8 assay determined the cell viability of primary hepatocytes following treatment with 5 mM glucose, 25 mM glucose, 25 mM glucose/fructose, or 25 mM glucose with 0.5 mM palmitic acid for 24 h. * *p* < 0.05 vs. 5 mM glucose group. # *p* < 0.05 vs. treatment-matched. All data are representative of at least three independent experiments and expressed as mean ± SEM. Charts were produced using GraphPad Prism 5.0. Statistical differences were determined by two-way ANOVA with Mann–Whitney *U* test using SPSS v17.0. WT, wild type; KO, knockout; CD, control chow diet; HSD, high-sucrose diet, CCK-8, Cell Counting Kit-8.

**Figure 6 ijms-22-00331-f006:**
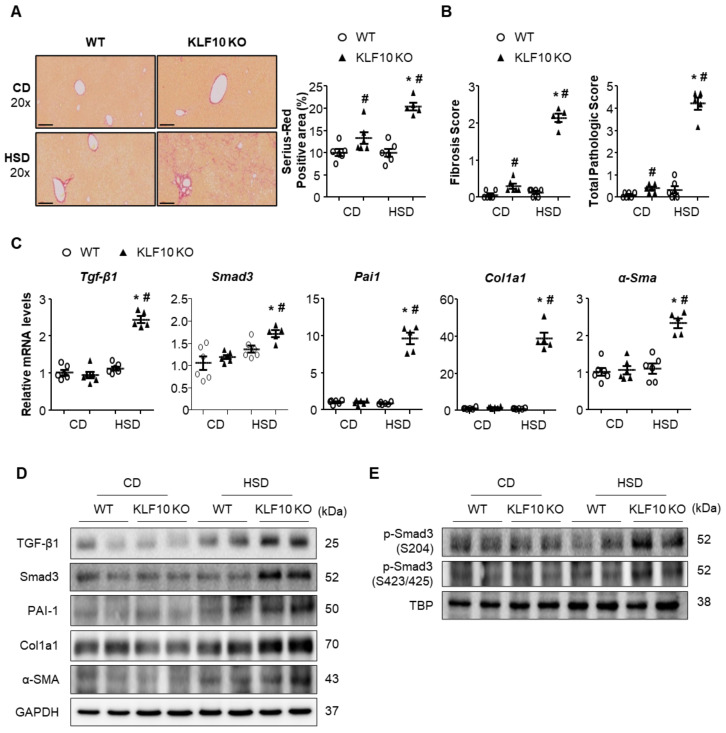
*Klf*10 deletion aggravates liver fibrosis through the activation of TGF-β-Smad3 pathway. Eight-week-old WT and *Klf*10 KO mice were fed either CD or HSD for eight weeks (n = 5–6 mice/group). (**A**) Collagen deposition was evaluated via Sirius Red staining of the liver sections (Scale bars: 60 μm, Magnification: 20×). Quantification indicates Sirius Red-positive area (percentage). (**B**) The grade of the fibrosis and NASH score was calculated. Changes in hepatic mRNA and protein expression involved in TGF-β (TGF-β1 and Smad3) and fibrogenesis (PAI-1, α-SMA, and Col1A1) pathway were analyzed via (**C**) qPCR and (**D**) Western blotting. (**E**) Nuclear extracts were used to detect the phosphorylation levels of Smad3 at S204 and S423/425. A TATA-binding protein (TBP) was used as a nuclear loading control. All data are representative of at least three independent experiments and expressed as mean ± SEM. Charts were produced using GraphPad Prism 5.0. Statistical differences were determined by two-way ANOVA with Mann–Whitney *U* test using SPSS v17.0. * *p* < 0.05 vs. genotype-matched, CD-fed group and # *p* < 0.05 vs. diet-matched, genotype control. WT, wild type; KO, knockout; CD, control chow diet; HSD, high-sucrose diet.

## Data Availability

The data presented in this study are available on request from the corresponding author.

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
