# Peer review of "Deletion of KLF10 Leads to Stress-Induced Liver Fibrosis upon High Sucrose Feeding"

_ijms, 2020, doi:10.3390/ijms22010331_

Round 1

Reviewer 1 Report

The authors in this work have successfully managed to prove and report their findings on the protective role of KFL10 against liver injury in and fibrosis in mice due to high sucrose intake.

The authors report herein that lack of KHF10 led to high sucrose related liver injury, impaired TG circulation and mitochondrial function. They also show that KFL10 affects hepatic cell death while its deletion leads to TGFB/SMAD3 related fibrosis.

The experiment is indeed well designed and executed covering answers to all generated questions in a step by step way leaving not much to question on their mechanistic findings.

Although the authors have designed and exexcuted a detailed set of experiments they seem to have issues expressing properly themselves in the introduction part.

  1. 1st Sentence: ´Liver fibrosis is a NAFLD´ is an expression that does is not correcr. Liver fibrosis is rather a consequence of NAFLD as well as other liver diseases. Please correct by rephrasing.
  2. 1st Sentence: NAFLD is one disease and not many. Not all liver diseases that are caused independently to alcohol consumption are called NAFLD.
  3. Intoduction: the authors state: NASH is characterized by liver steatosis, inflammation, hepatocellular injury, and varying degrees  of  fibrosis. This statement is partially correct. NASH is diagnosed when liver steatosis and inflammation occurs. Fibrosis could be also present but is not necessary to diagnose NASH.  More focused cited literature should be used.
  4. Intoduction: the authors state: Although KLF10  is  ubiquitously expressed, it is highly expressed in the Why is there an ´Although´ in the phrase?
  5. Introduction: The authors have written: Previous studies suggested that KLF10 may regulate liver pathophysiology in a context-dependent Citation for this phrase has to be added.
  6. Figure 1A. The weight axis should be presented with major units/splits of 5 instead of 10 g. Then the differences would be easier visualized.
  7. In all figure legends, the legend should include the software used for the production of charts and statistical analysis.
  8. In all figures, the western blots shown should have the kDa size indicated next to each marker tested.
  9. In all figures the microscopy images should include the magnification indicated close to the images.
  10. The authors talk of mitochondrial function. However no mitochondrial functional analysis has been performed (measurement of the oxygen consumption rate, mitochondrial membrane potential etc.) The authors should talk better for mitochondrial marker levels changes rather than mitochondrial function.
  11. In the end of their discussion the author state: Humans with genetic defects in Klf10 may undergo an accelerated transition from simple liver steatosis to NASH. That is a very strong hypothesis taking into consideration that no related human studies are being shown in this work. The sentence should be removed to avoid misleading the readers.
  12. Material and methods: Why have the mice fasted for 2 hr before euthanasia? A small sentence should be added with the fasting info on the results section.
  13. The findings are indeed interesting but whether this is also the case in humans still remains untouched. The authors could give us an insight with in vitro experiments on human cell lines of the above findings. If not possible related literature could possibly be added.

Kind Regards

Reviewer 2 Report

Lee and colleagues present very interesting data from a preclinical study investigating the effect of high sucrose feeding on liver fibrosis, reporting that deletion of KLF10 exacerbated the histology. Some concerns should be addressed:

Abstract:

Line 16-17. The sentence is incorrect. Liver fibrosis can be part of the spectrum of NAFLD, but is not necessarily so. I understand that the authors aim was to be synthetic, but the sentence has to be rephrased in order for it to maintain its accuracy.

Results:

I suggest to assess expression of genes involved in cholesterol excretion within the bile acids given the observed results (10.1007/s12020-019-02124-3)

Figure  1D: please provide a bigger image

Discussion:

Line 244-245 technically, I would rather say that its absence is associated with worse phenotype.I believe this concept should be implemented throughout the manuscript

FGF21 KO mice also develop fibrosis upon high sucrose, high fructose, and high fat feeding (10.1016/j.molmet.2018.03.002, 10.1016/j.molmet.2016.11.008, 10.1007/s12020-019-02124-3). The FGF21-CHREBP axis seems to be strongly involved. What do the authors think about it? It would be interesting to elaborate this aspect in relation to KLF10-CHREBP in the discussion section and measure FGf21 expression in these mice if at all possible.

Methods

The n number is very small and some of the results have a wide variability. it would be best to at least duplicate the n number for each group.

Round 2

Reviewer 2 Report

The authors have addressed all of my concerns and the manuscript has significantly improved.